# Plug and Play: Enabling Pluggable Attribute Unlearning in Recommender Systems

## Abstract

With the escalating privacy concerns in recommender systems, attribute unlearning has drawn widespread attention as an effective approach against attribute inference attacks. This approach focuses on unlearning users' privacy attributes to reduce the performance of attackers while preserving the overall effectiveness of recommendation. Current research attempts to achieve attribute unlearning through adversarial training and distribution alignment in the statistic setting. However, these methods often struggle in dynamic real-world environments, particularly when considering scenarios where unlearning requests are frequently updated. In this paper, we first identify three main challenges of current methods in dynamic environments, i.e., irreversible operation, low efficiency, and unsatisfied recommendation preservation. To overcome these challenges, we propose a Pluggable Attribute Unlearning framework, PAU. Upon receiving an unlearning request, PAU plugs an additional erasure module into the original model to achieve unlearning. This module can perform a reverse operation if the request is later withdrawn. To enhance the efficiency of unlearning, we introduce rate distortion theory and reduce the attack performance by maximizing the encoded bits required for users' embedding within the same class of the unlearned attribute and minimizing those for different classes, which eliminates the need to calculate the centroid distribution for alignment. We further preserve recommendation performance by constraining the compactness of the user embedding space around a reasonable flood level. Extensive experiments conducted on four real-world datasets and three mainstream recommendation models demonstrate the effectiveness of our proposed framework.

## CCS Concepts

• **Information systems** → **Recommender systems; Collaborative filtering**; • **Security and privacy** → **Social network security and privacy**.

## Keywords

Recommender Systems, Collaborative Filtering, Attribute Unlearning

**ACM Reference Format:**
Anonymous Author(s). 2018. Plug and Play: Enabling Pluggable Attribute Unlearning in Recommender Systems. In *Proceedings of Make sure to enter the correct conference title from your rights confirmation emai (Conference acronym 'XX)*. ACM, New York, NY, USA, 9 pages. https://doi.org/XXXXXXX.XXXXXXX

## 1 INTRODUCTION

Recommender systems employ highly personalized information extracted from user data to implement personalized recommendations, gaining widespread adoption in practical applications and profoundly influencing people's lifestyles [5, 13, 31]. However, as recommender systems continue to evolve, privacy concerns within personalized recommendations have increased, with more users demanding protection against the misuse of their sensitive information. As one protective measure, the *right to be forgotten* has been proposed [3, 30], requiring recommendation platforms to allow users to withdraw their personal data and its associated impacts. This legal requirement has spurred research on machine/recommendation unlearning.

Existing recommendation unlearning primarily focuses on input unlearning [4, 28], where the model inputs, such as users, items, or ratings, serve as the target for unlearning. Although input unlearning benefits multiple parties, it cannot effectively remove the latent attributes. This concern primarily stems from the attribute inference attacks (AIA) [1, 20], where the recommendation model, due to its powerful information extraction capabilities, might inadvertently encode sensitive attributes into its latent embeddings [1, 12]. This prompts research into attribute unlearning. User-sensitive attributes that are not directly used in training but are implicitly learned during the model embedding process [22] (e.g., gender, race, and age) serve as the target for attribute unlearning.

Existing research on attribute unlearning aims to reduce the effectiveness of attacks while preserving recommendation performance. These studies can be categorized into two types based on the stage of unlearning implementation [22]: in-training unlearning and post-training unlearning. In-training unlearning, such as adversarial training methods [1, 9], involves introducing an additional attack discriminator during model training, which can effectively maintain recommendation performance. However, this approach requires access to the original training data. Nevertheless, due to practical legal and regulatory constraints, accessing the original training data may be challenging during the execution of unlearning. In contrast, post-training unlearning, which involves manipulating model parameters without relying on training data, includes methods such as reducing attack performance through distribution alignment and maintaining recommendation performance through parameter regularization [6, 22]. Although post-training unlearning methods can avoid direct access to original training data, their application in dynamic real-world scenarios still faces challenges when unlearning requests frequently change.

For instance, modifications to privacy protection policies might lead to previously unlearned data attributes no longer requiring unlearning, or changes in the attributes themselves that need to be unlearned. Specifically, this approach encounters three main challenges: i) The unlearning operation is irreversible, as it involves altering model parameters; ii) The efficiency of unlearning is low, as each unlearning operation requires recalculating aligned centroid distributions; iii) The performance of recommendations is compromised, as frequent changes in unlearning requests significantly reduce the effectiveness of parameter regularization.

To address these challenges, we propose a novel pluggable attribute unlearning framework named PAU. Instead of altering the parameters of the original recommendation model, PAU framework introduces an additional erasure module. The user's embedding first passes through this erasure module before proceeding with the subsequent computation. Moreover, when there is a change in the unlearning request, only the erasure module needs to be replaced. In other words, our proposed erasure module features a plug-and-play design. Specifically, to optimize this erasure module, we need to achieve two key unlearning objectives. Firstly, to mitigate the potential impact of attacks, i.e., removing information related to the unlearned attributes from user embeddings, we aim for the users' embeddings from the same class of the unlearned attribute to be uncorrelated and to exhibit similarity to embeddings from other classes, thus making attackers difficult to extract attributes information. To enhance efficiency, we achieve this by maximizing the encoded bits (i.e., *rate distortion*) required for users' embedding within the same class of the unlearned attribute and minimizing those for different classes, which eliminates the need to calculate the centroid distribution for alignment. Further, to better preserve the recommendation performance, we use a rate-distortion function to limit the compactness of the user embedding space around a reasonable flood level, aimed at ensuring non-correlation with the unlearned attribute classes while minimizing the impact on other information, thereby reducing negative effects on recommendation performance. By combining these two objectives, we formulate a constrained optimization problem, and by solving this problem, we update and optimize the parameters of the erasure module. We summarize the main contributions of this paper as follows:

- We propose a pluggable recommendation attribute unlearning framework, featuring a plug-and-play design, which effectively addresses the frequently updated unlearning requests in the real world without altering the parameters of the original model.
- To enhance the efficiency of unlearning, we introduce rate-distortion theory, reducing the potential risk of attacks by maximizing the encoded bits required for users' embedding within the same unlearned attribute class and minimizing those for different attribute classes.
- To maintain the recommendation performance, we limit the compactness of the user embedding space to an adjustable flooding parameter through the rate distortion function, thereby minimizing the impact on other information outside the unlearned attributes.
- We conducted extensive experiments on four real-world datasets and three mainstream recommendation models. The experimental results validate the effectiveness of our method, and compared

to existing baselines, our method significantly outperforms them in terms of unlearning efficiency and maintaining recommendation performance.

## 2 RELATED WORK

### 2.1 Recommendation Model

Recommender systems provide personalized services based on the interaction information between users and items. Collaborative filtering (CF) is a widely recognized algorithm used to analyze such information [32], with its objectives comprising user and item embedding matrices. According to existing literature [21], CF is primarily categorized into three types: matrix factorization-based CF [26], neural network-based CF [17], and graph-based CF [16]. In this paper, we explore the problem of attribute unlearning within recommendation models, targeting user embeddings as the main focus for both attacks and unlearning, and we validate the generality of our methods across these three mainstream CF models.

### 2.2 Attribute Unlearning

Existing research on machine unlearning primarily focuses on unlearning specific samples from training data (i.e., input unlearning) while often overlooking potential attributes that are not explicitly represented in the training data. Guo et al. [12] are among the first researchers to explore the problem of attribute unlearning and proposed a method for unlearning specific attributes in facial images, such as smiles, beards, and large noses, through the manipulation of disentangled representations. Specifically, their approach decomposes the model into a feature extractor and a classifier and then inserts a network block between them to achieve manipulation. Additionally, Moon et al. [27] study attribute unlearning in generative models, including generative adversarial networks and Variational Autoencoders (VAE). They transform images containing the target attribute to images without it through unlearning.

### 2.3 Recommendation Attribute Unlearning

Given that recommendation systems capture sensitive user information such as gender, race, and age, the application of attribute unlearning in recommendation scenarios is particularly crucial. Ganhor et al. [9] employ an adversarial training approach using VAE to achieve attribute unlearning in recommendation models. This method implements unlearning during model training, involving manipulation of the training process and requiring access to the original data, which is often challenging to implement in real-world scenarios. In contrast, Li et al. [22] explore post-training attribute unlearning by directly manipulating model parameters without accessing training data or other training information such as gradients. They achieve unlearning through distribution alignment while maintaining recommendation performance through parameter regularization. However, both methods involve altering model parameters, which is unsuitable in real-world scenarios where unlearning requests frequently change. Thus, in this paper, we propose a pluggable attribute unlearning framework that incorporates an additional erasure module, allowing for attribute unlearning without altering the original parameters. This framework effectively addresses the challenges posed by dynamic environments in real-world scenarios.

# 3 PRELIMINARIES

## 3.1 Attacking Setting

Following the settings in previous research [22, 34, 37], the attack process in the attribute unlearning problem of recommender systems is also referred to as AIA, which is divided into three main stages: exposure, training, and inference. During the exposure phase, we adopt the assumption of a gray-box attack, meaning not all model parameters are exposed to the attacker; only the embeddings of the users and their related attribute information are revealed. In the training phase, it is assumed that the attacker trains the attacking model on an i.i.d. shadow dataset [29]. Although training using a shadow dataset might reduce the performance of the attack, such an assumption is reasonable because assuming that the attacker possesses the entire dataset is overly idealistic and impractical. The attack process is considered a classification task, where the attacking model takes user embeddings as input and attribute information as labels. In the inference phase, the attacker utilizes their attacking model to make predictions.

## 3.2 Rate Distortion

In our framework, the unlearning operation is implemented by ensuring non-correlation among user embeddings under the same unlearned attribute class, be consistent. Therefore, the attacker cannot infer sensitive attribute information through user embedding. To achieve this goal, an objective function known as rate distortion is employed to assess the compactness of a set of user embeddings. This section will elaborate on the fundamental principles of rate distortion theory [33, 36].

*Nonasymptotic rate distortion for finite samples.* In the field of lossy data compression [7], the compactness of random distribution is measured using rate distortion theory. For a given random variable $z$ and a prescribed precision $\epsilon$, the rate-distortion $R(z, \epsilon)$ is defined as the minimum number of binary bits required to encode $z$ to ensure that the expected decoding error is less than $\epsilon$, i.e., the decoded $\widehat{z}$ satisfies $\mathbb{E}\left[\|z - \widehat{z}\|_2\right] \leq \epsilon$. Although this framework successfully elucidates the feature selection mechanism in deep networks [24], a major challenge in practice is that we often do not precisely know the distribution of $z$, making $R(z, \epsilon)$ difficult to compute directly. Instead, we can only rely on a limited number of samples to learn representations. For example, given data samples $X = [x_1, \ldots, x_m]$, we have $z_i = f(x_i, \theta) \in \mathbb{R}^d, i = 1, \ldots, m$, where $m$ denotes the number of samples and $d$ denotes the dimension of the representation. Fortunately, prior research [23] has provided precise estimates of the binary bits required to encode finite samples from subspace distributions, with the total number of bits required given by the following expression:

$$\mathcal{L}(Z, \epsilon) = \left(\frac{m + d}{2}\right) \log \det \left(I + \frac{d}{m\epsilon^2} Z Z^\top\right), \quad (1)$$

where $Z = [z_1, \ldots, z_m]$ denotes the set of the learned representation. This enables us to apply this measure of compactness to real-world data, even though the underlying distribution of these data may not be well-defined. Further, the overall compactness of the learned features can be assessed by calculating the average encoding length per sample [23], i.e., the optimal encoding rate

w.r.t. precision $\epsilon$. Given that $m$ is usually much larger than $d$, the following expression can be derived:

$$R(Z, \epsilon) = \frac{1}{2} \log \det \left(I + \frac{d}{m\epsilon^2} Z Z^\top\right). \quad (2)$$

Generally, a set of compact vectors (low information content) requires fewer bits for encoding, corresponding to smaller values of $R(Z, \epsilon)$, and vice versa.

*Rate distortion of data with a mixed distribution.* In general, the feature set $Z$ of multi-class data may reside in multiple low-dimensional subspaces. To more accurately evaluate the rate distortion of such mixed data, it is feasible to partition data $Z$ into $k$ subsets, denoted as $Z = Z_1 \cup Z_2 \cup \ldots \cup Z_k$, where each subset lies within a distinct low-dimensional subspace. Consequently, the rate $R(Z_j, \epsilon)$ can be computed for the $j$-th subset utilizing the given formula Eq (2). To facilitate such computations, let $\Pi = \left\{\Pi_j \in \mathbb{R}^{m \times m}\right\}_{j=1}^k$ represent a set of diagonal matrices, where the diagonal elements signify the membership of $m$ samples in $k$ classes. More specifically, the diagonal element $\Pi_j(i, i)$ in matrix $\Pi_j$ indicates the probability that sample $i$ belongs to subset $j$. Thus, the matrix set $\Pi$ resides within a simplex space $\Omega = \left\{\Pi \mid \Pi_j \geq 0, \Pi_1 + \cdots + \Pi_k = I\right\}$. Following prior research [23], the average number of bits per sample (i.e., the rate distortion) w.r.t such a partition is:

$$R^c(Z, \epsilon \mid \Pi) = \sum_{j=1}^k \frac{\operatorname{tr}\left(\Pi_j\right)}{2m} \log \det \left(I + \frac{d}{\operatorname{tr}\left(\Pi_j\right) \epsilon^2} Z \Pi_j Z^\top\right). \quad (3)$$

When $Z$ is specified, $R^c(Z, \epsilon \mid \Pi)$ is a concave function of $\Pi$. The function $\log \det(\cdot)$ in the aforementioned expression has long been considered an effective heuristic for rank minimization problems, ensuring convergence to a local minimum [8].

# 4 METHODOLOGY

In this section, we first introduce our proposed pluggable attribute unlearning framework PAU, which is achieved by incorporating an erasure module. Then, we derive the two primary objectives of attribute unlearning, followed by the optimization process of the erasure module. Finally, we present the computational details.

## 4.1 Pluggable Attribute Unlearning Framework

To effectively manage the frequently updated unlearning requests encountered in real-world applications, we introduce an additional attribute erasure module, denoted as a function $f(\cdot)$. Figure 1 presents an overview of our proposed attribute unlearning framework. Specifically, when a set of user-item interaction data is input into the recommendation model, it first passes through the model's embedding layer, resulting in the corresponding user and item embedding matrices. We define the user embedding matrix as the representation set $U = U_1 \cup U_2 \cup \ldots \cup U_k$, where $k$ represents the number of class in the unlearned attribute (i.e., unlearning target), and $U^j$ denotes the embedding sub-matrix corresponding to the $j$-th class. The user embeddings first pass through the attribute erasure module, resulting in $\widehat{U}(\theta) = f(U, \theta) \in \mathbb{R}^{m \times d}$. For convenience, we will use $\widehat{U}$ to represent $\widehat{U}(\theta)$ in subsequent expressions. Then, $\widehat{U}$

### Collaborative filtering model

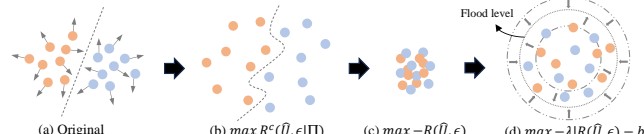

**Figure 1: Overview of pluggable attribute unlearning framework in recommender systems.**

replaces the original user embeddings $U$ and is used in conjunction with item embeddings for further computations in subsequent modules.

## 4.2 Erasure Module Optimization

To optimize the attribute erasure module, we primarily focus on two objectives in attribute unlearning: reducing the effectiveness of attacks and maintaining recommendation performance.

*Unlearning efficiency.* Previous research achieves these objectives through distribution alignment, typically involving two steps: first calculating the centroid distribution of all class distributions, followed by aligning the distributions. However, computing the centroid distribution is computationally expensive, and with each round of parameter update, the class distributions also change, necessitating the recalculation of the centroid distribution, which significantly limits unlearning efficiency. In light of this, we employ the rate distortion theory to link the attribute unlearning objectives with the erasure module. Thus, we can directly measure the compactness of all distributions without recalculation each round. In the remainder of this section, we conduct a detailed analysis of these two conceptual objectives to derive the formalized objectives.

*4.2.1 **Reducing the effectiveness of attacks**. To reduce the attacker's performance, i.e., to ensure that the correlations among user embeddings after processed by the erasure module are minimized, the transformed embeddings $\widehat{U}$ need to satisfy the following conditions:

- **Intra-class Inconsistency.** User embeddings belonging to the same class of the unlearned attribute should be highly uncorrelated and thus possess significant intra-class variance. This implies that each unlearned attribute class needs to cover a broader space (or subspace), thereby increasing the number of bits required to encode each sample within the class. Consequently, it is necessary to maximize $R^c(\widehat{U}, \epsilon \mid \Pi)$ .
- **Inter-class Consistency.** To reduce the distinguishability among user embeddings, the coherence between different unlearned attribute classes should be as high as possible. Consequently, these classes should collectively cover the smallest possible space, and the overall encoding rate of the set $\widehat{U}$ should be minimized, i.e., the rate distortion $R(\widehat{U}, \epsilon)$ should be minimized.

Formalizing these conditions can be expressed as:

$$\max_{\widehat{U},\Pi} \mathcal{J}\left(\widehat{U}, \Pi\right) = R^c\left(\widehat{U}, \epsilon \mid \Pi\right) - R(\widehat{U}, \epsilon). \tag{4}$$

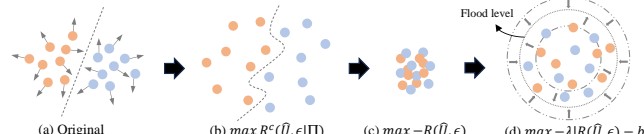

**Figure 2: Comparison of different optimization objectives.**

Considering that the objective function $\mathcal{J}(\widehat{U}, \Pi)$ is monotonic on the scale of user embeddings $\widehat{U}$ [36], and to ensure that the impact of different user embeddings on the objective function is comparable, it is necessary to normalize the scale of user embeddings after erasure. We achieve this by applying the Frobenius norm to scale user embeddings $\widehat{U}_j$ with the number of users: $\|\widehat{U}_j\|_F^2 = \sum_{i=1}^{m_j} \sum_{l=1}^{d} \widehat{U}_j(i, l)^2 = m_j$, where $m_j$ represents the number of samples in the class $j$.This is equivalent to normalizing each embedding to the unit sphere: $\widehat{U}_j(i, l) \in \mathbb{S}^{d-1}$. Thus providing a theoretical basis for implementing batch normalization in the practice of training deep neural networks [18]. Once the comparability of the impact of user embeddings on the objective function $\mathcal{J}(\widehat{U}, \Pi)$ is established, our objective can be written in the following form:

$$\max_{\widehat{U},\Pi} \mathcal{J}\left(\widehat{U}, \Pi\right) = R^c\left(\widehat{U}, \epsilon \mid \Pi\right) - R(\widehat{U}, \epsilon)$$

$$\text{s.t.} \quad \left\|\widehat{U}_j\right\|_F^2 = m_j, \Pi \in \Omega. \tag{5}$$

Solving Eq (5) can ensure that the transformed user embeddings are uncorrelated with the unlearned class information.

*4.2.2 **Maintaining recommendation performance.** Although our derived objective Eq (5) effectively reduces the effectiveness of attackers, it fails to maintain the performance of the recommender system and could even lead to a decline in recommendation performance. As illustrated in Figure 2 (b), when only the intra-class inconsistency condition is met, decision boundaries may still exist between different user embeddings. Based on this consideration, we introduce inter-class consistency conditions. However, as shown in Figure 2 (c), due to the user embedding space becoming extremely compact, a significant overlap of user embeddings occurs. This indeed results in the loss of the attribute information that needs to be unlearned but also leads to the loss of other useful information. Such destruction of general representational capacity ultimately causes a decline in recommendation performance.

To address this issue, a direct approach would be to relax the inter-class consistency constraints moderately. However, this relaxation needs to be limited; excessive relaxation might lead to a loss of the unlearning effect, potentially degrading to the state shown in Figure 2 (b) in extreme cases. Consequently, following [19], we adopt a solution called *flooding* as shown in Figure 2. When the overall embedding space becomes overly compact, specifically when $R(\widehat{U}, \epsilon)$ falls below a reasonable value $b$, which is referred to as *flood level*, we intentionally relax constraints. Conversely, when the embedding space is too loose, we proactively tighten the constraints. We ensure that $R(\widehat{U}, \epsilon)$ fluctuates around the flood level, thereby achieving a balance between the unlearning effect and maintaining recommendation performance. Specifically, we rewrite Eq (5) as:

$$\max_{\widehat{U}, \Pi} \widehat{\mathcal{J}}\left(\widehat{U}, \Pi\right) = R^c\left(\widehat{U}, \epsilon \mid \Pi\right) - \lambda \left|R(\widehat{U}, \epsilon) - b\right|$$

$$\text{s.t.} \quad \left\|\widehat{U}_j\right\|_F^2 = m_j, \Pi \in \Omega, \tag{6}$$

where $\lambda$ is a hyperparameter. In Eq (6), the second term represents the penalty introduced if the total volume of the embedding space (i.e., its compactness) deviates significantly from the parameter $b$.

## 4.3 Computational Implementation

To address the optimization problem posed by Eq (6), we employ an alternating optimization strategy [2, 11]. In each iteration, the variables $\Pi$ and $\widehat{U}$ are updated alternately. This strategy effectively decomposes the complex problem into manageable subproblems, thereby incrementally approaching the global optimum. We provide a detailed description below, and the complete algorithmic procedure can be found in Algorithm 1.

*Optimization of* $\Pi$. With $\widehat{U}$ held constant, we can directly compute the gradient of the objective function $\widehat{\mathcal{J}}(\cdot)$ w.r.t. $\Pi_j$ (i.e., element in $\Pi$). Given that $\Pi_j$ is a diagonal matrix, it suffices to derive the gradients for the diagonal elements, yielding:

$$\nabla_{\Pi_j} R^c(\widehat{U}, \epsilon \mid \Pi) = \frac{\text{tr}(\Pi_j)}{2m} \left(\widehat{U}^\top \left(I + \frac{d}{\text{tr}(\Pi_j) \epsilon^2} \widehat{U} \Pi_j \widehat{U}^\top\right)^{-1} \widehat{U}\right). \tag{7}$$

we can employ gradient ascent to update $\Pi_j$ to obtain $\Pi_j^{\text{new}}$. Since $\Pi$ must satisfy the constraint of simplex space: $\Pi \in \Omega$, it is projected onto set $\Omega$. Specifically, this involves successive projections for non-negativity $\Pi_j^{\text{proj}}(i, i) = \max(0, \Pi_j^{\text{new}}(i, i))$ and satisfying allocation constraints $\Pi_j^{\text{final}}(i, i) = \Pi_j^{\text{proj}}(i, i)/s_i$, where $s_i = \sum_{j=1}^k \Pi_j^{\text{proj}}(i, i)$.

*Optimization of* $\widehat{U}$. We first derive the gradient of $R(\widehat{U}, \epsilon)$ w.r.t. $\widehat{U}$. For convenience, we define $A = d/m\epsilon^2$, thereby allowing the rate distortion function to be rewritten as $R(\widehat{U}, \epsilon) = \frac{1}{2} \log \det \left(I + A\widehat{U}\widehat{U}^\top\right)$. Utilizing the properties of matrix differentiation, the gradient can be obtained as

$$\nabla_{\widehat{U}} R(\widehat{U}, \epsilon) = A\widehat{U} \left(I + \widehat{U}^\top A \widehat{U}\right)^{-1}. \tag{8}$$

Similarly, since $R^c(\widehat{U}, \epsilon \mid \Pi)$ is the sum of multiple determinants, a similar approach can be used to derive

$$\nabla_{\widehat{U}} R^c(\widehat{U}, \epsilon \mid \Pi) = \sum_{j=1}^k \frac{A\Pi_j \widehat{U}}{\text{tr}(\Pi_j)} \left(I + \frac{d}{\text{tr}(\Pi_j) \epsilon^2} \widehat{U} \Pi_j \widehat{U}^\top\right)^{-1}. \tag{9}$$

Then, we can employ gradient ascent to update $\widehat{U}$ to obtain $\widehat{U}^{\text{new}}$. Considering that each element of $\widehat{U}^{\text{new}}$ must satisfy the constraint of Frobenius norm, the following scaling operation is required to project it onto the set that satisfies the constraint: $\widehat{U}_j^{\text{final}}(\theta) = \widehat{U}_j^{\text{new}}(\theta) \cdot \sqrt{m_j / \|\widehat{U}_j^{\text{new}}(\theta)\|_F^2}$.

---

**Algorithm 1** Pluggable Attribute Unlearning

---

1: **Input:** User embedding $U$, training epoch $E$, adjustable threshold $b$, hyperparameter $\lambda$, update step size $\eta$.
2: **Initial:** Erasure module parameter $\theta_0$, erased user embedding $\widehat{U}(\theta_0) = \widehat{U_1}(\theta_0) \cup \widehat{U_2}(\theta_0) \cup \ldots \cup \widehat{U_k}(\theta_0)$, set of diagonal matrices $\Pi^0 = \{\Pi_j^0 \in \mathbb{R}^{m \times m}\}_{j=1}^k$.
3: **for** $e = 0$ to $E$ **do**
4:     Compute gradients: $\nabla_{\Pi_j^e} R^c(\widehat{U}, \epsilon \mid \Pi)$ as defined in Eq (7).
5:     Update $\Pi^{e+1}$: $\Pi_j^{e+1} \leftarrow \Pi_j^e + \eta \nabla_{\Pi_j^e} R^c(\widehat{U}, \epsilon \mid \Pi)$.
6:     Non-negative projection: $\Pi_j^{e+1} = \max(0, \Pi_j^{e+1}(i, i))$.
7:     Allocation projection: $\Pi_j^{e+1} = \Pi_j^{e+1}(i, i)/\sum_{g=1}^k \Pi_g^{e+1}(i, i)$.
8:     Compute gradients: $\nabla_{\widehat{U}(\theta_e)} \widehat{\mathcal{J}}(\widehat{U}(\theta_e), \Pi)$ as defined in Eq (8) and Eq (9).
9:     Updata parameter: $\widehat{U}(\theta_{e+1}) = \widehat{U}(\theta_e) + \eta \nabla_{\widehat{U}(\theta_e)} \widehat{\mathcal{J}}(\widehat{U}(\theta_e), \Pi)$.
10:     Frobenius norm projection: $\widehat{U}_j(\theta_{e+1}) = \widehat{U}_j(\theta_{e+1}) \cdot \sqrt{m_j / \|\widehat{U}_j(\theta_{e+1})\|_F^2}$.
11: **end for**
12: **return** Erasure module parameter $\theta_E$.

---

## 5 EXPERIMENTS

In this section, we evaluate three aspects, i.e., unlearning effectiveness, recommendation performance, and unlearning efficiency. We conduct unlearning on various attributes, including both binary and multi-class attributes. Additionally, we investigate the effect of hyper-parameters, and conduct an ablation study for each component of our proposed loss function.

### 5.1 Experimental Settings

*5.1.1 **Datasets**.* We conduct experiments on four publicly accessible real-world datasets, each containing user-item interaction data (i.e., ratings) and user attribute data (e.g., gender and age).

- **MovieLens 100K (ML-100K):** The MovieLens dataset is recognized as one of the most extensively utilized resources for recommendation systems research [14, 15]. It encompasses user ratings for movies along with various attributes such as gender, age, and occupation. ML-100K specifically comprises approximately 100,000 interaction records.
- **MovieLens 1M (ML-1M):** ML-1M comprises approximately 1 million records, providing a broader scope for analysis.

**Table 1: Summary of datasets.**

| Dataset | Attribute | Category # | User # | Item # | Rating # | Sparsity |
|---|---|---|---|---|---|---|
| ML-100K | Gender
Age | 2
3 | 943 | 1,349 | 99,287 | 92.195% |
| ML-1M | Gender
Age | 2
3 | 6,040 | 3,416 | 999,611 | 95.155% |
| LFM-2B | Gender
Age | 2
3 | 19,972 | 99,639 | 2,829,503 | 99.858% |
| KuaiSAR | Feat1
Feat2 | 8
2 | 21,852 | 140,367 | 2,166,893 | 99.929% |

- **LFM-2B**: THe LFM-2B dataset includes over 2 billion listening events designed for music retrieval and recommendation purposes [25]. LFM-2B also collects users' attributes such as gender, age, and country. In our experiments, we utilize a subset that contains more than 3 million interaction records.
- **KuaiSAR**[1]: KuaiSAR serves as a comprehensive dataset for recommendation search, derived from user behavior logs collected from the short-video mobile application, Kuaishou[2]. In our experiments, we utilize KuaiSAR-small, which includes two anonymous user attributes (i.e., Feat1 and Feat2).

We summarize the statistics of the above datasets in Table 1.

*Dataset pre-processing.* For these datasets, we first exclude users with incomplete or invalid attribute information. Next, we retain only users who have interacted with at least five items and items that have received at least five user interactions.

*Train-test split.* To assess recommendation performance, the two most recent interaction items from each user (sorted by interaction timestamp) are retained, one for validation and the other for testing.

*Dataset attributes.* The age attribute is divided into three groups: under 35 years old, between 35 and 45, and over 45. The available gender attribute is restricted to male and female categories. For KuaiSAR, we use anonymized one-hot encoded categories of users as the target attributes.

*5.1.2* **Recommendation Models.** We validate the effectiveness of our proposed method across three well-acknowledged recommendation models.

- **DMF**: Deep Matrix Factorization (DMF) is a prominent deep learning model within the framework of matrix factorization [35].
- **NCF**: Neural Collaborative Filtering (NCF) is a foundational collaborative filtering model that employs neural network architectures [17].
- **LightGCN**: Light Graph Convolution Network (LightGCN) is the State-Of-The-Art (SOTA) collaborative filtering model that optimizes recommendation performance through a simplified graph convolutional network design [16].

*Training parameters.* For model-specific hyper-parameters in recommendation models, we adhere to the recommendations provided in respective original papers. Specifically, we utilize the SGD

---

[1]https://kuaisar.github.io/
[2]https://www.kuaishou.com/

optimizer with a learning rate of 1e-4 and set the embedding dimension to 32. All model parameters are initialized using a Gaussian distribution $\mathcal{N}(0, 0.1^2)$

*5.1.3* **Evaluation metrics.** We specify the evaluation metrics of unleanring effectiveness and recommendation performance as follows.

*Unlearning Effectiveness.* As mentioned in Section 3.1, the attacking process is considered a classification task, where the attacking model takes user embeddings as input and attributes information as labels. Following [22], We build a Multilayer Perceptron (MLP) [10] as an adversarial classifier, because MLP stands out as the attacker with the best performance. The dimension of MLP's hidden layer is set as 100 and a softmax layer is used as the output layer. we set the L2 regularization weight to 1.0, the initial learning rate to 1e-3, and the maximal iteration to 500, leaving the other hyper-parameters at their defaults in scikit-learn 1.1.3. To evaluate the effectiveness of attribute unlearning, we utilize two widely used classification metrics: the micro-averaged F1 score (F1) and Balanced Accuracy (BAcc). Lower values of F1 and BAcc indicate greater effectiveness of unlearning. We train the MLP using 80% of users and test with the remaining 20%. We report the results of attacking through five-fold cross-validation, averaged over 10 runs.

*Recommendation Effectiveness.* To assess the recommendation performance, we employ the leave-one-out testing. We leverage Hit Ratio at rank K (HR@K) and Normalized Discounted Cumulative Gain at rank K (NDCG@K) as measures of recommendation performance. HR@K measures whether the test item is in the top-K list, while NDCG@K is a position-aware ranking metric that assigns higher scores to the hits at the upper ranks. In our experiment, the entire negative item sets are used to compute HR@K and NDCG@K. Note that we compare the recommendation effectiveness performance of several compared methods under the condition of achieving optimal unlearning effectiveness.

*5.1.4* **Unlearning Methods.** We compare our proposed method (PAU) with the original model and three representative attribute unlearning methods.

- **Original**: This is the original model without attribute unlearning.
- **DP [38]**: This method protects user attributes by introducing noise perturbation to the user embedding during the model prediction process.
- **AU [22]**: This method represents the SOTA attribute unlearning method, which is achieved through distribution alignment.
- **Adv [9]**: This method uses adversarial training to achieve attribute unlearning. Thus, it not only intervenes in the training process of the original model, but also requires access to the training data.

We run all models 10 times and report the average results.

*Hyper-parameters.* To obtain the optimal performance, we use grid search to tune the hyper-parameters. The number of epochs is set to 15 for DMF and NCF, and 200 for LightGCN. Additionally, in our proposed PAU, we assign the trade-off coefficient as $\lambda = 1$ and set the flood level $b$ to 3.

**Table 2: Results of unlearning performance (i.e., the performance of attackers, denoted by pink blocks) and recommendation performance (denoted by yellow blocks). Except for Original, the best results are highlighted in bold.**

| Dataset | Attribute | Method | DMF | | | | | | NCF | | | | | | LightGCN | | | | | |
|---|---|---|---|---|---|---|---|---|---|---|---|---|---|---|---|---|---|---|---|---|
| | | | HR@5 | NDCG@5 | HR@10 | NDCG@10 | BAcc | F1 | HR@5 | NDCG@5 | HR@10 | NDCG@10 | BAcc | F1 | HR@5 | NDCG@5 | HR@10 | NDCG@10 | BAcc | F1 |
| ML-100K | Gender | Original | 0.0961 | 0.0688 | 0.1550 | 0.0830 | 0.6889 | 0.6746 | 0.1023 | 0.0674 | 0.1610 | 0.0850 | 0.6771 | 0.6639 | 0.1074 | 0.0678 | 0.1697 | 0.0880 | 0.6000 | 0.6145 |
| | | DP | 0.0130 | 0.0090 | 0.0210 | 0.0110 | 0.5149 | 0.4660 | 0.0310 | 0.0200 | 0.0540 | 0.0270 | 0.6586 | 0.6665 | 0.0300 | 0.0190 | 0.0580 | 0.0290 | 0.5906 | 0.5476 |
| | | AU | 0.0940 | 0.0600 | **0.1530** | 0.0790 | 0.4712 | 0.3962 | 0.0980 | **0.0650** | 0.1530 | **0.0830** | 0.4529 | 0.4104 | **0.1043** | 0.0665 | 0.1659 | **0.0854** | 0.5113 | 0.5287 |
| | | Adv | 0.0820 | 0.0520 | 0.1470 | 0.0790 | 0.6881 | 0.6742 | 0.0840 | 0.0540 | 0.1510 | 0.0740 | 0.5673 | 0.5517 | 0.1006 | 0.0644 | 0.1524 | 0.0812 | 0.5401 | 0.5517 |
| | | PAU (ours) | **0.0947** | **0.0671** | 0.1528 | **0.0822** | 0.4539 | 0.3865 | 0.1018 | 0.0645 | **0.1607** | 0.0821 | 0.4016 | 0.3967 | 0.1033 | **0.0672** | 0.1676 | 0.0843 | 0.5164 | 0.5340 |
| | Age | Original | 0.0961 | 0.0680 | 0.1550 | 0.0830 | 0.6660 | 0.6746 | 0.1023 | 0.0674 | 0.1610 | 0.0850 | 0.5607 | 0.5609 | 0.1074 | 0.0678 | 0.1697 | 0.0880 | 0.5102 | 0.6025 |
| | | DP | 0.0130 | 0.0090 | 0.0210 | 0.0110 | 0.3810 | 0.3809 | 0.0310 | 0.0200 | 0.0540 | 0.0270 | 0.5133 | 0.5138 | 0.0300 | 0.0190 | 0.0580 | 0.0290 | 0.5126 | 0.5126 |
| | | AU | 0.0830 | 0.0520 | 0.1430 | **0.0710** | 0.2830 | 0.2824 | 0.1000 | **0.0640** | 0.1540 | 0.0810 | 0.2061 | 0.2062 | 0.0975 | 0.0625 | 0.1556 | 0.0792 | 0.3443 | 0.5710 |
| | | Adv | 0.0680 | 0.0450 | 0.1180 | 0.0600 | 0.5989 | 0.6017 | 0.0950 | 0.0620 | 0.1530 | **0.0820** | 0.3761 | 0.5974 | 0.1006 | 0.0651 | 0.1581 | 0.0845 | 0.3688 | 0.6047 |
| | | PAU (ours) | **0.0895** | **0.0549** | **0.1488** | 0.0708 | 0.2651 | 0.2742 | **0.1007** | 0.0638 | **0.1542** | 0.0818 | 0.2354 | 0.2529 | **0.1016** | **0.0658** | **0.1597** | **0.0853** | 0.3177 | 0.5067 |
| ML-1M | Gender | Original | 0.0658 | 0.0401 | 0.1060 | 0.0520 | 0.7487 | 0.7501 | 0.0699 | 0.0439 | 0.1126 | 0.0589 | 0.7485 | 0.7545 | 0.0690 | 0.0430 | 0.1100 | 0.0579 | 0.7022 | 0.6987 |
| | | DP | 0.0090 | 0.0050 | 0.0150 | 0.0070 | 0.7059 | 0.7023 | 0.0090 | 0.0050 | 0.0150 | 0.0070 | 0.7341 | 0.7318 | 0.0260 | 0.0160 | 0.0430 | 0.0220 | 0.7080 | 0.7055 |
| | | AU | 0.0600 | 0.0380 | 0.1000 | **0.0510** | 0.4910 | 0.4868 | 0.0660 | 0.0410 | 0.1050 | 0.0530 | 0.4504 | 0.4662 | **0.0664** | 0.0421 | 0.1087 | 0.0559 | 0.5068 | 0.5187 |
| | | Adv | 0.0560 | 0.0360 | 0.0970 | 0.0500 | 0.6106 | 0.6181 | 0.0470 | 0.0280 | 0.0980 | 0.0480 | 0.5551 | 0.5574 | 0.0634 | 0.0397 | 0.1035 | 0.0532 | 0.5515 | 0.5874 |
| | | PAU (ours) | **0.0642** | **0.0397** | **0.1038** | 0.0502 | 0.4396 | 0.4317 | **0.0688** | **0.0437** | **0.1115** | **0.0574** | 0.4485 | 0.4693 | 0.0661 | **0.0425** | **0.1098** | **0.0565** | 0.4819 | 0.4939 |
| | Age | Original | 0.0658 | 0.0401 | 0.1060 | 0.0520 | 0.7487 | 0.7501 | 0.0699 | 0.0439 | 0.1126 | 0.0589 | 0.6241 | 0.6241 | 0.0690 | 0.0430 | 0.1100 | 0.0579 | 0.5664 | 0.5664 |
| | | DP | 0.0090 | 0.0050 | 0.0150 | 0.0070 | 0.5150 | 0.5150 | 0.0090 | 0.0050 | 0.0150 | 0.0070 | 0.6110 | 0.6110 | 0.0260 | 0.0160 | 0.0430 | 0.0220 | 0.5626 | 0.5625 |
| | | AU | 0.0600 | 0.0380 | 0.0990 | 0.0500 | 0.2911 | 0.2911 | 0.0650 | 0.0400 | **0.1040** | 0.0530 | 0.2611 | 0.2611 | **0.0669** | **0.0422** | 0.1077 | 0.0556 | 0.3347 | 0.5671 |
| | | Adv | 0.0500 | 0.0320 | 0.0880 | 0.0450 | 0.7120 | 0.7120 | 0.0510 | 0.0320 | 0.0990 | 0.0490 | 0.3707 | 0.6125 | 0.0621 | 0.0382 | 0.1058 | 0.0528 | 0.3779 | 0.6114 |
| | | PAU (ours) | **0.0679** | **0.0395** | **0.1003** | **0.0543** | 0.2475 | 0.2491 | **0.0691** | **0.0436** | 0.1038 | **0.0567** | 0.2367 | 0.2722 | 0.0667 | 0.0421 | **0.1088** | **0.0562** | 0.3180 | 0.5228 |
| LFM-2B | Gender | Original | 0.0120 | 0.0080 | 0.0200 | 0.0100 | 0.6802 | 0.6717 | 0.0170 | 0.0110 | 0.0280 | 0.0140 | 0.6779 | 0.6809 | 0.0220 | 0.0140 | 0.0340 | 0.0180 | 0.6162 | 0.6219 |
| | | DP | 0.0010 | 0.0000 | 0.0030 | 0.0010 | 0.6349 | 0.6278 | 0.0010 | 0.0010 | 0.0020 | 0.0010 | 0.6691 | 0.6727 | 0.0030 | 0.0020 | 0.0060 | 0.0030 | 0.6187 | 0.6179 |
| | | AU | 0.0050 | 0.0030 | 0.0080 | 0.0040 | 0.4478 | 0.4477 | 0.0120 | 0.0070 | 0.0210 | 0.0107 | 0.4470 | 0.4527 | 0.0176 | 0.0102 | 0.0271 | **0.0145** | 0.5032 | 0.5114 |
| | | Adv | **0.0100** | 0.0070 | 0.0160 | **0.0073** | 0.6259 | 0.6336 | 0.0130 | 0.0080 | 0.0210 | 0.0100 | 0.5436 | 0.5547 | 0.0165 | 0.0098 | 0.0260 | 0.0135 | 0.5479 | 0.5643 |
| | | PAU (ours) | 0.0095 | **0.0072** | **0.0169** | 0.0072 | 0.3876 | 0.3911 | **0.0153** | **0.0090** | **0.0263** | **0.0124** | 0.4125 | 0.4172 | 0.0175 | **0.0103** | **0.0274** | 0.0144 | 0.4682 | 0.4965 |
| | Age | Original | 0.0120 | 0.0080 | 0.0200 | 0.0100 | 0.6802 | 0.6717 | 0.0170 | 0.0110 | 0.0280 | 0.0140 | 0.3187 | 0.3787 | 0.0220 | 0.0140 | 0.0340 | 0.0180 | 0.3353 | 0.3351 |
| | | DP | 0.0010 | 0.0000 | 0.0030 | 0.0010 | 0.3320 | 0.3323 | 0.0010 | 0.0010 | 0.0020 | 0.0010 | 0.3325 | 0.3328 | 0.0030 | 0.0020 | 0.0060 | 0.0030 | 0.3295 | 0.3293 |
| | | AU | **0.0120** | 0.0070 | **0.0190** | 0.0090 | 0.3330 | 0.3331 | 0.0120 | 0.0070 | 0.0200 | 0.0100 | 0.3160 | 0.3160 | **0.0220** | 0.0140 | 0.0350 | **0.0180** | 0.3274 | 0.3275 |
| | | Adv | 0.0080 | 0.0050 | 0.0160 | 0.0080 | 0.3624 | 0.3626 | 0.0150 | **0.0090** | 0.0220 | 0.0110 | 0.3750 | 0.3745 | 0.0140 | 0.0090 | 0.0240 | 0.0120 | 0.3246 | 0.3248 |
| | | PAU (ours) | 0.0116 | **0.0077** | 0.0173 | **0.0104** | 0.3141 | 0.3014 | 0.0141 | 0.0086 | **0.0233** | **0.0115** | 0.2261 | 0.2232 | 0.0216 | **0.0142** | **0.0358** | 0.0176 | 0.2259 | 0.2236 |
| KuaiSAR | Feat1 | Original | 0.0190 | 0.0120 | 0.0310 | 0.0150 | 0.2667 | 0.2483 | 0.0180 | 0.0126 | 0.0370 | 0.0178 | 0.3427 | 0.4152 | 0.0200 | 0.0130 | 0.0360 | 0.0180 | 0.2132 | 0.4419 |
| | | DP | 0.0030 | 0.0020 | 0.0050 | 0.0020 | 0.1787 | 0.1809 | 0.0020 | 0.0010 | 0.0050 | 0.0020 | 0.4333 | 0.4242 | 0.0060 | 0.0040 | 0.0110 | 0.0050 | 0.2812 | 0.2866 |
| | | AU | 0.0150 | 0.0100 | 0.0270 | 0.0130 | 0.1167 | 0.1135 | 0.0180 | **0.0120** | 0.0300 | 0.0150 | 0.1652 | 0.3760 | 0.0193 | 0.0127 | 0.0328 | 0.0173 | 0.1426 | 0.3819 |
| | | Adv | **0.0170** | 0.0100 | **0.0310** | **0.0150** | 0.2305 | 0.2253 | 0.0150 | 0.0100 | **0.0350** | 0.0170 | 0.1608 | 0.4065 | 0.0195 | 0.0124 | 0.0317 | 0.0165 | 0.1681 | 0.4125 |
| | | PAU (ours) | 0.0168 | **0.0117** | 0.0306 | 0.0129 | 0.1019 | 0.1077 | **0.0185** | 0.0116 | 0.0313 | **0.0172** | 0.1541 | 0.2475 | **0.0195** | **0.0128** | **0.0331** | **0.0175** | 0.1140 | 0.3357 |
| | Feat2 | Original | 0.0190 | 0.0120 | 0.0310 | 0.0150 | 0.2667 | 0.2483 | 0.0180 | 0.0126 | 0.0370 | 0.0178 | 0.1500 | 0.1522 | 0.0200 | 0.0130 | 0.0360 | 0.0180 | 0.7481 | 0.7488 |
| | | DP | 0.0030 | 0.0020 | 0.0050 | 0.0020 | 0.5333 | 0.5333 | 0.0020 | 0.0010 | 0.0050 | 0.0020 | 0.1590 | 0.1549 | 0.0060 | 0.0040 | 0.0110 | 0.0050 | 0.5777 | 0.5503 |
| | | AU | 0.0150 | 0.0090 | 0.0270 | 0.0130 | 0.1759 | 0.1715 | 0.0180 | 0.0110 | 0.0300 | 0.0150 | 0.1311 | 0.1309 | 0.0186 | **0.0125** | 0.0331 | 0.0168 | 0.5476 | 0.5543 |
| | | Adv | 0.0160 | **0.0100** | 0.0300 | **0.0150** | 0.6333 | 0.6327 | 0.0080 | 0.0050 | **0.0350** | **0.0170** | 0.2143 | 0.2251 | 0.0185 | 0.0124 | 0.0324 | 0.0164 | 0.5821 | 0.5957 |
| | | PAU (ours) | **0.0172** | 0.0096 | **0.0313** | 0.0149 | 0.1542 | 0.1571 | **0.0182** | **0.0119** | 0.0325 | 0.0159 | 0.1016 | 0.1181 | **0.0192** | 0.0123 | **0.0341** | **0.0173** | 0.5242 | 0.5471 |

*Hardware information.* All models and algorithms are implemented using Python 3.8 and PyTorch 1.9. The experiments are conducted on a server running Ubuntu 20.04, equipped with 256GB of RAM and an NVIDIA GeForce RTX 4090 GPU.

## 5.2 Results and Discussion

### 5.2.1 Unlearning Effectiveness.
Reducing the attacking performance of AIA is the primary goal of attribute unlearning. To comprehensively evaluate attacking performance, we report two metrics, i.e., F1 score and BAcc, in Table 2. We have the following observations from the above results.

- Firstly, attackers achieve an average F1 score of 0.54 and BAcc of 0.56 on the original embedding, indicating the users' attribute information in embeddings can be inferred by the attacker. Note that the average performance of these results encompasses both binary and multi-class attributes. Thus, a value of 0.5 does not represent the optimal outcome.
- Secondly, all compared methods can effectively unlearn attribute information in embeddings to varying degrees. DP, AU, Adv, PAU decrease the F1 scores by an average of 14.23%, 35.37%, 11.76%, and 40.72%, respectively. Furthermore, PAU achieves an average reduction in BAcc by 43.32%, while AU only reduces BAcc by 37.90%. Thus, our proposed PAU demonstrates a significant advantage over compared unleanring methods.

### 5.2.2 Recommendation Performance.
Preserving recommendation performance is another important goal of attribute unlearning. While users' attributes are unlearned, the impact on recommendation performance should be minimized to promise the utility of the recommendation model. We use NDCG and HR to evaluate recommendation performance after unlearning and truncate the rank list at 5 and 10 for both metrics. As shown in Table 2 unlearning methods indeed affect recommendation performance in varying degrees. DP significantly decreases the utility of the recommendation model. Specifically, compared with the original model, DP decreases the NDCG@5 and NDCG@10 by 83.00% and 80.02%, and decreases HR@5 and HR@10 by 83.83% and 78.97%, respectively, on average. Adv and AU decrease NDCG by 27.57% and 22.57%, and decrease the HR by 29.39% and 24.41%, respectively, on average. As for comparison, PAU only has an average degradation of 19.53% on NDCG and 17.17% on HR. This can be attributed to the fact that PAU more effectively preserves recommendation performance by incorporating the flooding-level controlled compactness regularization.

### 5.2.3 Unlearning Efficiency.
We use running time to assess the efficiency of unlearning methods. We conduct experiments on ML-1M dataset and report the running time of both binary (gender) and multi-class (age) attribute unlearning. From Figure 3, we observe that

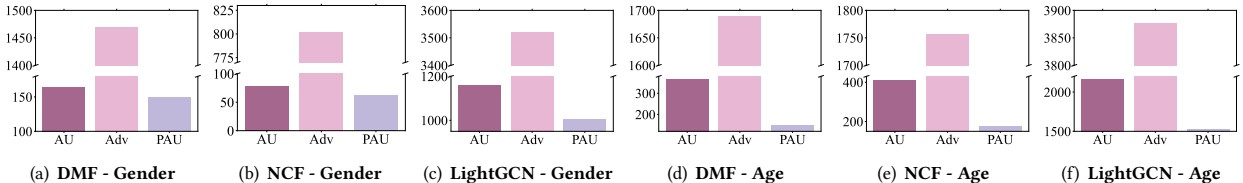

(a) **DMF - Gender**  (b) **NCF - Gender**  (c) **LightGCN - Gender**  (d) **DMF - Age**  (e) **NCF - Age**  (f) **LightGCN - Age**

**Figure 3: Results of unlearning efficiency. We present the running time of compared methods on ML-1M dataset across three recommendation models. The results are reported in seconds (s).**

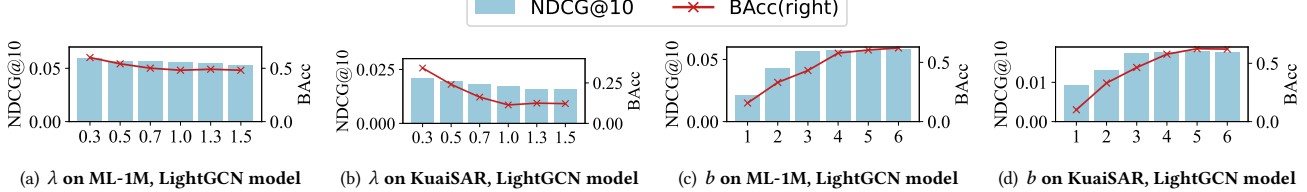

(a) $\lambda$ **on ML-1M, LightGCN model**  (b) $\lambda$ **on KuaiSAR, LightGCN model**  (c) $b$ **on ML-1M, LightGCN model**  (d) $b$ **on KuaiSAR, LightGCN model**

**Figure 4: Effect of the hyper-parameter trade-off coefficient $\lambda$ and flood level $b$. We conduct experiments on ML-1M dataset and KuaiSAR dataset, using the LightGCN model. We use BAcc and NDCG@10 to represent the performance of unlearning and recommendation respectively.**

- Our proposed PAU significantly outperforms Adv. Compared to Adv, PAU reduces the running time by 90.49%, 90.73%, and 65.73% on DMF, NCF, and LightGCN respectively. This is because Adv uses a time-consuming adversarial training approach to achieve attribute unlearning.
- Compared to AU, PAU reduces the running time of unlearning binary attributes by 13.39% on average.
- For the multi-class attribute unlearning, PAU demonstrates a more significant advantage, achieving an average reduction of 36.96%. This is because AU spends exponentially more time re-calculating centroid distribution for multi-class attributes. In contrast, our proposed PAU utilizes rate distortion theory to directly measure the compactness of distributions, thereby significantly enhancing unlearning efficiency.

*5.2.4 Parameter Sensitivity Analysis.* We investigate the effect of key hyper-parameters, i.e., trade-off coefficient $\lambda$ and flood level $b$, on both unlearning effectiveness and recommendation performance. As shown in Fig 4, we use BAcc and NDCG@10 to represent the performance of unlearning and recommendation respectively.

- **Trade-off Coefficient $\lambda$.** Regarding the trade-off coefficient $\lambda$, we set $\lambda$ values to 0.3, 0.5, 0.7, 1.0, 1.3, 1.5. We observe insignificant fluctuation of $\lambda$. The results demonstrate that our PAU method's BAcc and NDCG@10 exhibit good robustness to different $\lambda$ values.
- **Flood Level $b$.** Concerning the flood level $b$, we compare model performance with flood level $b$ values of 1, 2, 3, 4, 5, and 6. We observe that as $b$ increases, both NDCG and BAcc. But the fluctuations are not significant. After $b = 3$, the increase levels off, forming a plateau. To better preserve the recommendation performance, we set $\lambda = 1$ for other experiments. Note that the maximal BAcc remains below 0.6.

## 6 CONCLUSIONS

This paper investigates the problem of attribute unlearning in recommender systems, aiming to protect user attribute information from attackers while maintaining recommendation performance. Existing methods employ adversarial training and distribution alignment to update model parameters for attribute unlearning. However, these methods are challenging to apply in dynamic real-world environments, particularly when unlearning requests are frequently updated. The primary challenges faced by these methods include irreversible operation, low efficiency, and unsatisfied recommendation preservation. To overcome these challenges, we propose a pluggable attribute unlearning framework, PAU, based on rate distortion theory. Our proposed framework reduces attack performance by maximizing the bits required to encode user embeddings within the same unlearned attribute class and minimizing those for different attribute classes. Additionally, PAU maintains recommendation performance by constraining the compactness of the user embedding space around a reasonable flood level. We conducted extensive experiments on four real-world datasets and three mainstream recommendation models to evaluate the effectiveness of our proposed method. The results demonstrate that our approach effectively achieves attribute unlearning under pluggable conditions and significantly outperforms existing baseline methods in terms of unlearning efficiency and maintaining recommendation performance. Our research offers valuable insights for other machine unlearning tasks, encouraging future studies to focus more on practical scenarios. In future research, we plan to explore the unlearning of continuous attributes or more generic vector attributes, which may require further improvements and optimization of our current methods.

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

Received 20 February 2007; revised 12 March 2009; accepted 5 June 2009

