# OpenReview forum: "Plug and Play: Enabling Pluggable Attribute Unlearning in Recommender Systems"
_ACM.org/TheWebConf/2025/Conference — WWW 2025 Poster_

### Official Review · Reviewer_Ve49 · 2024-11-28

**Novelty:** 3
**Technical Quality:** 2

**Review:**

The paper addresses a relevant problem—attribute unlearning in recommender systems—and provides a detailed explanation of its proposed solution, the Pluggable Attribute Unlearning (PAU) framework.  The use of rate-distortion theory is a theoretically sound idea, but the integration and explanation lack depth. Experimental evaluations cover several datasets and baseline models, but the choice of baselines is not very convincing, and the results are not sufficiently compelling.

**Questions:**

1. The paper lacks comparisons with more recent and relevant baselines in the field of machine unlearning in recommender systems (RS). For instance, [1] ("Post-Training Attribute Unlearning in Recommender Systems") and [2] ("Making Recommender Systems Forget: Learning and Unlearning for Erasable Recommendation") both use the same outdated baselines as this paper. Since these works represent the current state-of-the-art, it is necessary to compare the proposed method with them to demonstrate its effectiveness.


2. Several interesting phenomena in Table 2 are not adequately discussed in the paper. For example, the NDCG scores for PAU outperform the original model for DMF, which is unexpected and counterintuitive. This suggests that the proposed framework may have inadvertently improved the recommendation performance beyond simply unlearning sensitive attributes. At the same time, the BAcc and F1 metrics show significant improvements, indicating effective unlearning. This raises an apparent contradiction: how does the framework improve recommendation quality while effectively erasing sensitive attribute information?

3. The metrics used to evaluate unlearning quality, such as BAcc and F1, are insufficient to fully capture the quality of unlearning. Recent literature, such as the survey in [3] ("A Survey of Machine Unlearning"), suggests alternative metrics that could provide a more comprehensive evaluation. Metrics such as Completeness, Retraining Time, Layer-Wise Distance, and JS-Divergence should be considered.

4. The second contribution, which claims the introduction of rate-distortion theory into the framework, is overstated. The authors do not introduce or develop rate-distortion theory but rather employ it as an optimization tool within the erasure module. While this is a valid application, the motivation for using rate-distortion theory is neither well-justified nor convincing. The authors fail to explain why this specific theory is necessary for this context and how it offers advantages over alternative approaches.

5. Given that the paper is titled "Plug-and-Play" and targets the broader domain of recommender systems, the experiments are overly restricted to relatively simple methods like Matrix Factorization (MF) and Collaborative Filtering (CF). Modern RS techniques, such as Transformer-based models (SASRec, BERT4Rec), are conspicuously absent. These advanced models often capture complex patterns and richer representations, which may interact differently with the proposed framework.

6. The datasets used in this study, while standard, are relatively small and may not fully capture the challenges of attribute unlearning in large-scale, real-world systems. Larger datasets, such as Yelp, should be used.

7. The paper lacks critical implementation details, such as the specific architecture of the models used, hyperparameter settings. The readers will hard to re-produce the results.

8. The paper does not provide any statistical treatment of the experimental results. Improvements observed in Table 2 may be due to differences in hyperparameter tuning rather than true architectural novelty or methodological improvements.

9. Any investigation on why the dimension of MLP’s hidden layer is set as 100? We normally use 128 or 256. 100 seems a very magic number here.

**Reviewer Confidence:**

3: The reviewer is confident but not certain that the evaluation is correct

**Scope:**

3: The work is somewhat relevant to the Web and to the track, and is of narrow interest to a sub-community

---

### Official Review · Reviewer_fW3J · 2024-12-01

**Novelty:** 6
**Technical Quality:** 6

**Review:**

# Evaluation

**Quality**
This paper addresses the critical and emerging issue of attribute unlearning in recommender systems by introducing the Pluggable Attribute Unlearning (PAU) framework. The framework’s modular design allows for reversible and efficient unlearning operations without altering the base model parameters, which is highly practical for real-world applications.

**Clarity**
The paper is generally well-written, with clear motivation, methodology, and results. The use of diagrams and detailed experimental evaluations enhances comprehension.

**Originality**
The introduction of a plug-and-play erasure module, combined with rate-distortion theory for unlearning efficiency, is novel and addresses significant limitations in existing methods. The framework's ability to handle dynamic unlearning requests is a standout contribution.

**Significance**
The work tackles a vital problem in privacy-preserving recommender systems, offering a solution that balances unlearning effectiveness, recommendation performance, and efficiency. If widely adopted, this framework could set new standards for privacy protection in recommendation systems.

---

# Pros
- **Innovative Framework**: The plug-and-play design is a practical solution for dynamic unlearning requirements.
- **Technical Rigor**: The use of rate-distortion theory to optimize the embedding space is mathematically sound.
- **Comprehensive Experiments**: Results across multiple datasets and models showcase the framework's generalizability and effectiveness.

---

# Cons
- **Lack of Ablation Analysis**: While the authors mentioned an ablation study in Section 5, no comprehensive analysis of the individual components of PAU (e.g., the erasure module) is provided.

- **Missing Statistical Significance Test**: The claim that PAU significantly outperforms Adv in Section 5.2.3 lacks statistical validation, such as a paired t-test or other significance tests.

**Questions:**

# Questions for Authors

- Could you please provide more details on the ablation analysis? Specifically, how do individual components of PAU, such as the erasure module or the rate-distortion-based optimization, contribute to its performance?
- Would you mind including statistical significance tests (e.g., t-tests) to validate claims of PAU's superiority over baselines?

**Reviewer Confidence:**

3: The reviewer is confident but not certain that the evaluation is correct

**Scope:**

4: The work is relevant to the Web and to the track, and is of broad interest to the community

---

### Official Review · Reviewer_qj9B · 2024-12-01

**Novelty:** 6
**Technical Quality:** 6

**Review:**

In this paper, the authors introduce the Pluggable Attribute Unlearning (PAU) framework, which facilitates the removal of specific data from recommendation systems. I recommend accepting this paper.

The paper addresses a novel and timely topic. Machine unlearning is becoming increasingly important, as poisoned data can compromise the accuracy and effectiveness of trained models. Such data must be removed. Additionally, the authors highlight an essential perspective for machine unlearning: privacy concerns. Attackers may exploit models to extract sensitive attribute information, which makes unlearning critical, particularly for widely used recommendation systems in applications like e-commerce and short video platforms.

The authors used real-world datasets, which is a strong choice. All four datasets include attributes such as gender and age, which were utilized in the unlearning process.

Table 2 effectively showcases PAU’s unlearning performance in comparison to three other unlearning methods. The results clearly demonstrate that PAU outperforms the alternatives.

Moreover, the authors addressed parameter-related concerns in Section 5.2.4, which clarified issues raised in Section 5.1.4.

However, I have a few questions:

a. In Section 5.2.3, is there a specific reason why the authors used only the ML-1M dataset to evaluate efficiency?
b. Why did the comparison focus solely on AU and ADV against PAU? Why was DP excluded?

Adding explanations for these choices would strengthen the paper.

Despite these minor concerns, I find this paper valuable and strongly recommend its acceptance.

**Questions:**

a. In Section 5.2.3, is there a specific reason why the authors used only the ML-1M dataset to evaluate efficiency?
b. Why did the comparison focus solely on AU and ADV against PAU? Why was DP excluded?

**Reviewer Confidence:**

3: The reviewer is confident but not certain that the evaluation is correct

**Scope:**

4: The work is relevant to the Web and to the track, and is of broad interest to the community

---

### Official Review · Reviewer_aRzg · 2024-12-02

**Novelty:** 5
**Technical Quality:** 5

**Review:**

Attribute unlearning has gained significant attention for its role in protecting user privacy. However, existing research on attribute unlearning faces challenges in scenarios where unlearning requests are frequently updated, such as irreversible operations, low efficiency, and unsatisfactory recommendation preservation. To address these issues, the authors propose a pluggable attribute unlearning model, which optimizes both learning efficiency and recommendation performance.

Overall, this paper is well-structured, with comprehensive experiments demonstrating some improvement in resisting attribute inference attacks. However, the authors need to clarify the necessity of their proposed innovation.


Strengths:
(1) The introduction of distortion rate theory in the Erasure Module improves learning efficiency.
(2) The proposed method achieves a balance between privacy protection and recommendation performance, which appears reasonable.
(3) The comparative experiments are relatively comprehensive and provide adequate validation.

Weaknesses:
(1) In the “Methodology” section, the manuscript lacks the theoretical analysis explaining why these particular objective functions were chosen.
(2) In the Parameter Sensitivity Analysis, the authors should provide a deeper analysis of how parameter variations specifically affect the results rather than merely describing observed trends.
(3) The manuscript does not clearly define scenarios where unlearning requests are frequently updated, or justify whether such scenarios occur frequently enough to be significant.

**Questions:**

(1) What makes attribute inference attacks particularly threatening to user privacy compared to other attack methods? Why is this specific type of attack emphasized in this paper?
(2) The authors state, "This module can perform a reverse operation if the request is later withdrawn." Could the authors clarify what exactly constitutes a reverse operation in this context? How is this reversibility achieved?

**Reviewer Confidence:**

3: The reviewer is confident but not certain that the evaluation is correct

**Scope:**

3: The work is somewhat relevant to the Web and to the track, and is of narrow interest to a sub-community